# Modification of Thin Film Composite Pressure Retarded Osmosis Membrane by Polyethylene Glycol with Different Molecular Weights

**DOI:** 10.3390/membranes12030282

**Published:** 2022-02-28

**Authors:** Siti Nur Amirah Idris, Nora Jullok, Woei Jye Lau, Akmal Hadi Ma’Radzi, Hui Lin Ong, Muhammad Mahyidin Ramli, Cheng-Di Dong

**Affiliations:** 1Faculty of Chemical Engineering and Technology, Universiti Malaysia Perlis, Kompleks Pusat Pengajian Jejawi 3, Kawasan Perindustrian Jejawi, Arau 02600, Perlis, Malaysia; nuramie@gmail.com (S.N.A.I.); akmalhadi@unimap.edu.my (A.H.M.); hlong@unimap.edu.my (H.L.O.); 2Centre of Excellence for Biomass Utilization & Taiwan-Malaysia Innovation Centre for Clean Water and Sustainable Energy (WISE Center), Universiti Malaysia Perlis, Lot 17, Kompleks Pusat Pengajian Jejawi 2, Jejawi, Arau 02600, Perlis, Malaysia; 3Advanced Membrane Technology Research Centre, Universiti Teknologi Malaysia—UTM, Skudai 81310, Johor, Malaysia; lwoeijye@utm.my; 4Faculty of Electronic Engineering Technology, Universiti Malaysia Perlis, Changlun—Kuala Perlis Highway, Arau 02600, Perlis, Malaysia; mmahyiddin@unimap.edu.my; 5Department of Marine Environmental Engineering, National Kaohsiung University of Science and Technology, 142, Hai-Chuan Road, Nan-Tzu District, Kaohsiung 81157, Taiwan; cddong@nkust.edu.tw

**Keywords:** pressure retarded osmosis, flat sheet thin-film composite membrane, structural parameter, internal concentration polarization, power density

## Abstract

An investigation of the effect of the molecular weight of polyethylene glycol (PEG) on thin-film composite (TFC) flat sheet polysulfone membrane performance was conducted systematically, for application in forward osmosis (FO) and pressure retarded osmosis (PRO). The TFC flat sheet PSf-modified membranes were prepared via a non-solvent phase-separation technique by introducing PEGs of different molecular weights into the dope solution. The TFC flat sheet PSf-PEG membranes were characterized by SEM, FTIR and AFM. The PSf membrane modified with PEG 600 was found to have the optimum composition. Under FO mode, this modified membrane had a water permeability of 12.30 Lm^−2^h^−1^ and a power density of 2.22 Wm^−2^, under a pressure of 8 bar in PRO mode, using 1 M NaCl and deionized water as the draw and feed solutions, respectively. The high water permeability and good mechanical stability of the modified TFC flat sheet PSF-PEG membrane in this study suggests that this membrane has great potential in future osmotically powered generation systems.

## 1. Introduction

The growth of global energy demands and the rise in carbon emissions have led to an active exploration of renewable energy sources, such as solar [1], wind [2], geothermal [3] and biofuel [4] energy. The availability of most renewable energy sources is dependent, in some manner, on the day, season, year, or geographical location. Salinity gradient power (SGE) (or osmotic power) is a renewable energy source that is currently under the spotlight due to its potential in power generation [5,6,7,8,9]. Generally, the SGE membrane processes, such as forward osmosis (FO) and pressure retarded osmosis (PRO), are driven osmotically. These membranes operate according to the osmotic pressure difference between solutions on each side of the membrane (i.e., low salinity (LS) and high salinity (HS)).

In FO, water flows naturally from LS to HS via a semipermeable membrane without applying any pressure to the system (∆P = 0) (Figure 1). For the PRO process, the osmotic pressure (∆P < ∆π) is applied on the HS side, which partially retards the water’s movement across the semipermeable membrane [10], thus allowing the water to flow towards the HS solution. Unlike the FO membrane, a PRO membrane requires sufficient mechanical strength to withstand the high hydraulic pressure being applied.

In PRO, a semipermeable membrane is used to separate the HS and LS solutions by setting the orientation of the active layer to face the draw solution (AL-DS), as shown in Figure 2. Water diffuses from the feed side with low salinity (FS-LS) into the pressurized draw solution containing the high salinity (DS-HS) solution. The pressurized and diluted draw solution then moves through a depressurizing hydro turbine to produce power [11].

In membrane processes such as PRO, the membrane substrate layer plays an important role in the process, as it acts as the backbone for withstanding the high pressure applied to the system. Basically, polysulfone (PSf) is the most commonly used polymer in PRO membranes. It possesses good mechanical, thermal and chemical stabilities [12]. In addition, the high solubility of PSf makes it an ideal selection for polymer-blend membranes since other polymers or additives can be co-dissolved with it, thus enhancing the membrane’s performance. Many researchers have achieved good results when using PSf as a substrate [13,14,15,16,17,18]. However, the major problem with this polymer is its hydrophobic properties, which cause resistance to water adsorbing on the surface of the membrane and eventually lead to reduced membrane performance [13]. Therefore, modifications to improve the intrinsic properties of the pristine membrane are necessary. Such modifications can be accomplished by setting up a solvent/non-solvent system and by the manipulation of variables; for example, selecting a polymer or polymer concentration, or modifying the composition of the coagulation bath and composition of the casting solution.

The manipulation of the composition of a dope solution containing an additive, for instance, may assist in enhancing the pore formation or improving the pore connectivity of the membrane structure and introduce hydrophilicity [5,6,7,8,9,19,20]. The most commonly used additives for polymeric membranes include polyvinylpyrrolidone (PVP), polyethylene oxide (PEO), polyethylene glycol (PEG), inorganic salts (such as zeolite, titania (TiO_2_), alumina (Al_2_O_3_) and lithium chloride (LiCl)) and organic compounds (for example, alcohols, di-alcohols, glycerol and water). Among these additives, PEG is a well-known macro-void suppressor and can render the membrane hydrophilic. Ma et al. [21] incorporated different molecular weights (M_W_) of PEG in their flat sheet asymmetric PSf ultrafiltration membrane, and the results showed that the pure water flux (PWF) and porosity of the resulting membranes increased with the increase in the M_W_ of the PEG. However, the incorporation of the highest dosage, i.e., PEG 1500, tended to negatively affect the membrane’s mechanical properties. Zhou et al. [12] modified a PSf-FO substrate by blending it with methoxypolyethylene glycol, together with different M_W_ of PEG (200, 500, 1000 and 1900 gmol^−1^). Their work demonstrated that the water fluxes of the modified membranes increased dramatically when a PEG with a M_W_ of 500 gmol^−1^ was used to modify the PSf substrate. Sharma et al. [9] used different M_W_s of PEG as additives in their flat cellulose acetate asymmetric membrane, and found that the power density of the membrane was enhanced at increased M_W_s of the PEG due to its improved hydrophilicity, PWF, salt rejection, and porosity.

The limited work reported in the literature on the effects of adding PEG with various M_W_s to TFC flat sheet PSf membranes for FO and PRO applications has become the motivation of this study. It was anticipated that the performance of FO and PRO will improve with the modified TFC membrane, as the addition of PEG on the PSf substrate will form the hydrophilic fine-pore membrane structure. This study is a systematic investigation into the PEG-modified PSf substrates produced via a non-solvent-induced phase-separation followed by an interfacial polymerisation of the PA layer on the top surface of the membrane. The transitions of the membrane morphology, structure and their performances were tested through FO and PRO experiments.

## 2. Materials and Methods

### 2.1. Chemicals and Materials

Polysulfone granules (PSf, Udel^®^ P-1700 LCD, M_w_ 67,000 gmol^−1^, Solvay, Princeton, NJ, USA), 1-methyl-2-pyrrolidone (NMP, 99% M_w_ 99.13 gmol^−1^, Merck, Darmstadt, Germany) and polyethylene glycol (PEG; Merck, Germany) (M_w_: 400 gmol^−1^, 600 gmol^−1^, 1000 gmol^−1^, 1500 gmol^−1^, 4000 gmol^−1^ and 6000 gmol^−1^) were used to prepare the membrane substrates. 1,4-Phenylenediamine (PPD, 98% purity, M_w_ 108.14 gmol^−1^, Merck, Germany), 1,3,5-benzenetricarbonyl trichloride (TMC, 98% purity, M_w_ 265.47 gmol^−1^, Merck, Germany) and n-hexane (95% purity, M_w_ 86.18 gmol^−1^, Merck, Germany) were used in the interfacial polymerization. Sodium chloride (NaCl, M_w_ 58.44 gmol^−1^, HmbG^®^ Chemicals, Hamburg, Germany) was used in the preparation of the draw solution.

### 2.2. Synthesis of the Membranes

The polymer substrates were prepared via the non-solvent-induced phase-separation (NIPS) method. PSf was dissolved in NMP, and a certain amount of PEG 400 or PEG with a different M_w_ was added to the solution. The casting solution was mixed using an IKA^®^ Roller 6 Basic until it became homogeneous. Table 1 shows the composition of the casting solutions used for membrane synthesis. The dope solution was spread on a glass plate using an automated casting machine. The homogenous dope solution was evenly cast across the glass plate at a controlled thickness of 200 µm using a stainless-steel casting knife. The glass plate was then dipped straightway into a non-solvent bath containing DI water for 5 min to allow the phase-inversion immersion precipitation process to take place. The polymeric film (i.e., nascent membrane) was then kept under running DI water for 5 min to remove the residual solvent, and stored in DI water before the interfacial polymerization was carried out.

The polyamide (PA) active layer on the substrate was formed via interfacial polymerization (IP). The top surface of the support layer was soaked in a 2 wt% PPD solution for 5 min. After drying at room temperature for 30 s, a 0.15 wt% of TMC in n-hexane was poured onto the support layer for 1 min. Then the membrane was dried in a vacuum oven at 60 °C for 30 min. The prepared TFC flat sheet membrane was rinsed thoroughly with DI water to remove the residual monomers, and kept in the DI water until it was used.

### 2.3. Membrane Characterizations

#### 2.3.1. Morphology, Surface Roughness and Functional Group

The morphologies of the cross-sections and active layers of the membranes were observed via scanning electron microscopy (SEM, JEOL JSM-6460LA) operated at 10 kV and 10 mA. The membranes were first dried in the oven for 24 h to remove moisture. The membranes were then fractured under liquid nitrogen to obtain consistent and clean cuts of the cross-sectional areas. All the samples were sputter coated with platinum prior to observation.

Atomic force microscopy (AFM, SPA400-SP13800, Seiko Instruments Inc., Chiba, Japan) was used to analyse the surface roughness of the membranes and to render three-dimensional images of their surfaces. Small parts of the membranes, approximately 1 cm^2^, were cut and glued on glass substrates. Then, the membranes were imaged in tapping mode at resonance frequencies of 117 kHz at a 5 μm scan rate.

The functional group of membranes was also examined via Fourier-transform infrared spectroscopy (FTIR) using the attenuated total reflectance (ATR) mode (Spectrum 65, Perkin Elmer, Waltham, MA, USA). The spectrum for each sample was scanned 32 times from 450 cm^−1^ to 4000 cm^−1^ spectral regions with 4 cm^−1^ resolutions.

#### 2.3.2. Equilibrium Water Content, Porosity, Average Pore Size and Contact Angle

The equilibrium water content (*EWC*) and porosity (*ε*) of the membranes were measured via the differences between wet and dry weights. The membranes stored in DI water were weighed after being mopped with tissue paper using an electronic balance. Then, the wet membranes were dried in a vacuum oven for 24 h at a temperature of 60 °C and weighed in the dry state. The *EWC* and porosity were calculated with Equations (1) and (2):(1)$$EWC=\frac{m_{w}-m_{d}}{m_{w}}\times 100$$
(2)$$\varepsilon =\frac{m_{w}-m_{d}}{\rho \times T\times A_{m}}$$
where *m_w_* is the wet weight, *m_d_* is the dry weight, *ρ* is the density of DI water, *T* is the thickness, and *A_m_* is the effective area of the membrane sample.

The average pore size, *r_m_* was determined by using the pure water permeability *PWP* and porosity data in the Gerout–Elford–Ferry equation (Equation (3)) [22]:(3)$$r_{m}=\sqrt{8\times \frac{\left(2.9-1.75\varepsilon \right)\times \eta \times T\times PWP}{\varepsilon }}$$
where *η* and *T* are the viscosity of DI water and the thickness of the membrane sample, respectively. The *PWP* of the membranes was determined using a Sterlitech HP4750 high pressure stirred cell at 1 bar, with an effective membrane area of 14.6 cm^2^.

The contact angle (CA) of the membranes were measured with a contact angle instrument (OCA 15Pro, DataPhysics). Ten measurements were carried out at random locations on each membrane’s active layer to yield an average CA value for that membrane.

#### 2.3.3. RO Test for Intrinsic Parameters

The intrinsic parameters of the membranes were evaluated with the same high pressure stirred cell used to find the *PWP*. The water permeability, *A*, was measured using DI water as a feed solution and pressurized at 10 bar. The value of *A* was calculated using Equation (4):(4)$$A=\frac{\Delta V_{a}}{\Delta t_{a}\times A_{m}\times \Delta P}$$
where Δ*V_a_* is the permeate volume, Δ*t_a_* is the predetermined time, *A_m_* is the effective area of the membrane sample, and Δ*P* is the transmembrane pressure difference.

The salt permeability, *B*, was measured using 1000 ppm of NaCl as a feed solution under a hydraulic pressure of 10 bar. The salt rejection, *R*, was calculated via the following equation:(5)$$R=\left(1-\frac{C_{p}}{C_{f}}\right)\times 100\%$$
where *C_f_* and *C_p_* are the permeate and feed salt concentrations, respectively. The value of *B* was calculated using Equation (6) below, where Δ*P* and Δπ are the transmembrane hydraulic and osmotic pressure differences, respectively.
(6)$$B=A\times \frac{1}{R}\times \left(\Delta P-\Delta \pi \right)$$

The structural parameter, *S*, of the membrane was calculated via the classical flux-fitting method (Equation (7)), where *D* is the solute diffusion coefficient; π*_draw_* and π*_feed_* represent the osmotic pressures of the draw and feed solutions, respectively; and *J_w_* is the flux under the FO mode [22,23].
(7)$$S=\frac{D}{J_{w}}\ln\left[\frac{B+\left(A\times \pi_{draw}\right)}{B+J_{w}+\left(A\times \pi_{feed}\right)}\right]$$

### 2.4. FO and PRO Membrane Performances

The FO performance was evaluated using a cross-flow filtration setup with an effective membrane area of 0.0042 m^2^. The draw solution (DS) and feed solution (FS) containing 1 M NaCl and DI water, respectively, were circulated co-currently through a membrane cell at a fixed flow rate of 1.5 L/min. The weight changes in the DS for the FO, and the FS for the PRO, were recorded by a data-logging balance every 2 s for 5 h after the system was stabilized. The NaCl concentration changes in the feed and draw solutions were recorded with a conductivity meter (Lovibond, Sensor Direct 150). The FO performance was tested with the membrane active layer facing the feed solution (AL-FS), while the PRO performance was tested with the active layer facing the draw solution (AL-DS). The pressure on the draw solution in PRO was controlled in the range of 2 to 8 bar, and all the experiments were carried out at 24 ± 1 °C.

The water flux, *J_w_*, was determined by using the volume change in the DS (FO) and FS (PRO), Δ*V*, in a predetermined time interval, Δ*t*, in Equation (8):(8)$$J_{w}=\frac{\Delta V}{A_{m}\times \Delta t}$$
where *A_m_* is the effective membrane area of the membrane.

The salt reverse-flux, *J_s_*, was calculated using Equation (9):(9)$$J_{s}=\frac{C_{t}\times V_{t}-C_{0}\times V_{0}}{A_{m}\times \Delta_{t}}$$
where *C_t_* and *C*_0_ are the initial and final concentrations of the feed solution, respectively, and *V_t_* and *V*_0_ are the initial and final volumes of the feed solution, respectively.

The power density, *W* was calculated as the product of the water flux, *J_w_*, and the applied operating pressure, Δ*P*.
(10)$$W=J_{w}\times \Delta P$$

## 3. Results and Discussions

### 3.1. Membrane Morphology, Surface Roughness and Functional Group

Figure 3 shows the SEM images of the cross-sections and top surfaces of the pristine TFC membrane and modified TFC membranes. Based on the cross-section images, all the synthesized membranes exhibit typical asymmetric morphologies that consist of numerous finger-like pores separated by a sponge-like porous medium. The pristine PSf dope solution had fewer demixing conditions compared to the PSf-PEG dope solution, and increased the rate of the phase inversion, resulting in the pristine PSf membrane producing small, finger-like pores and lower membrane flux. However, when PEG was added with increasing M_w,_ there was a noticeable difference in the morphologies of the membranes due to the dissolution of the PEG, which consumed some of the solvent and resulted in higher dope solution viscosity. The dope solution became thermodynamically less stable as it tended to undergo rapid, instantaneous demixing when immersed in the coagulation bath. As the M_w_ of the PEG increased, the finger-like pores became larger and more irregular, indicating that hydrophilic PEG was beneficial as a pore-forming agent, resulting in the improved porosity of the membranes.

The membrane morphology change was expected and seen as being derived from the slower phase-inversion rate [22]. At the same time, the PSf-PEG solution had more time to relax and develop the polymer-lean-phase growth of the pores, resulting in the larger finger-like pores [24]. It is believed that the looser and more porous morphology minimizes the effect of the internal concentration polarization (ICP) in the FO/PRO process, and allows the homogenous formation of the polyamide active layer on top of the membrane surface [22,25]. Basically, PEG is considered to be a weak non-solvent for PSf-NMP solutions; thus, increasing the M_w_ of the PEG causes the dope solution to become thermodynamically less stable, and simultaneously increases the flow rate of the water across the membrane due to its intrinsic hydrophilicity [26].

The PA active layer was synthesized on top of the PSf membrane via an IP process between TMC and PPD. The SEM images of the top surface-active layers of the membranes are shown in Figure 3. All the PA active layers of the membranes exhibit a typical nodular-like structure. Hence, the successful crosslinking between TMC and PPD (Figure 4) created the desired conditions for the formation of a stable, thin film on the membrane surface area [27]. However, there are obvious differences between the pristine TFC membrane and modified TFC membranes, due to the formation of thin layers with higher surface roughness and grain-like structures formed on the membrane with PEG. As the M_w_ of the PEG increased, smooth, globule, spherical-like structures were observed on the top surface of the modified TFC membrane (Figure 3), which were potentially due to the pore size differences among the membranes, i.e., higher membrane porosity tended to increase the immersed PPD solution’s reactions with the TMC molecules. As a result, the membranes with higher PEG M_w_s exhibited thicker PA layers.

Figure 5 presents the 3D AFM micrographs of the top surface topologies of the pristine TFC membrane and modified TFC membranes. The average surface roughness (*R_avg_*) of the membranes decreased with the increasing M_w_ of the PEG. According to the data tabulated in Table 2, the *R_avg_* value for TFC-600 was 62.43 ± 3.01 nm, and it was 37.17 ± 2.74 nm for TFC-6000. It is postulated that this phenomenon was due to the presence of the crosslinking reaction in the IP process and the lack of free amide groups able to assist in reducing the surface roughness of the PA layer [29]. The *R_avg_*, maximum peak-to-valley distance (*R_p-v_*) and root-mean-squared roughness (*R_rms_*) for each membrane are presented in Table 2.

The functional groups of the membranes were characterized by FTIR, and the results are shown in Figure 6. According to Yi et al. [30], PEG with the chemical structure of (HO-CH_2_-(CH_2_-O-CH_2_)n-CH_2_-OH) shows significant peaks at 1016 cm^−1^, 1153 cm^−1^ and 1107 cm^−1^ that are attributed to the stretching of ether groups. The spectrum also shows a broad transmittance band for O-H (from the COOH group) stretching at 3333–3500 cm^−1^. PEG has the unique ability to dissolve in both aqueous and organic solvents, as it has both hydrophilic and hydrophobic properties. A similar finding signifying the presence of PEG in homogenously blended PSf in NMP solution was reported by Yunos et al. [26], namely, peaks at 2873 cm^−1^ and 3061 cm^−1^. The strong peaks at 1488 cm^−1^ and 1588.5 cm^−1^, which represent the amide-II aromatic in-plane ring C-H bending, indicate the successful formation of the PA layer [31,32]. Furthermore, other PA transmittance peaks were observed at 698.5 cm^−1^, 719.5 cm^−1^ and 837.5 cm^−1^, which can be attributed to the aromatic C=C stretching and C-O stretching of the carboxylic acids groups, which were formed by hydrolysis of the un-reacted acid chloride groups of TMC in the crosslinking PA process [33,34]. This characteristic of the synthesized membranes contributed to the increased PWP of the modified TFC membranes, which will be discussed in the following section.

### 3.2. Effects of PEG on the Intrinsic Properties of the TFC PA/PSf Membranes

Water permeability (*A*), salt permeability (*B*) and salt rejection (*R*) are the key indicators used to signify the optimization of membrane performance [35]. The effects of PEG 400 dosage on *A*, *B* and *R* for TFC PA/PSf-400 membranes are summarized in Table 3. This PEG with low M_w_, i.e., PEG 400, produced a stable dope solution. Zhou et al. [12] modified a PSf substrate by blending it with methoxypolyethylene glycol, together with different M_W_s of PEG. The water fluxes of the modified membranes increased dramatically when a PEG with a M_W_ of 500 gmol^−1^ was used. As the PEG concentration increases, the ratio of solvent to polymer increases and affects the membrane formation process [21,36]. Therefore, it is important to study the effect of wt% PEG 400 on the intrinsic properties of modified TFC membrane, which has been used as the benchmark in this study.

The performance of the synthesized membranes—based on the *A*, *B* and *R* values—improved with the increase in the pore-forming PEG 400 dosage, since the 10 wt% of PEG in TFC-400d had the highest values for both *A* and *R,* recorded as 1.19 Lm^−2^h^−1^bar^−1^ and 88% rejection, respectively. On the other hand, the pristine TFC-PSf membrane had the lowest *A* value of 0.49 Lm^−2^h^−1^bar^−1^ and rejected just 68% of the salt. Based on the results shown in Table 3, the TFC-400d membrane was chosen as the basis for investigating the effect of different M_w_s of PEG on the intrinsic hydrodynamic behaviours of the PSf/PEG membranes. The 10 wt% of PEG was fixed as the optimum PEG dosage, similar to other studies found in the literature [25,26,36,37]. 

Figure 7 shows the intrinsic transport properties of the modified TFC membrane with different M_w_s of PEG. The *A* value obviously increased as the M_w_ of the PEG increased. Comparing the membranes with the lowest and the highest M_w_s of PEG, the *A* value for TFC-6000 shows a 37% improvement over that for the TFC-400d membrane. Meanwhile, the TFC-600 has the highest *R* value among all the modified membranes, with 88% rejection. Table 4 shows the comparison of the *A*, *B* and *R* values of TFC-600 with other membranes modified with PEG-600 found in the literature.

It is postulated that this improvement was due to the construction of the active PA layer, which became smoother and looser, with a globular-like structure, with increases in the M_w_ of the PEG. Furthermore, the *B* value increased slightly with the increase in the M_w_ of the PEG, but the *R* value decreased [22]. According to Equation (6), the *B* value has a positive correlation with the *A* value but a negative correlation with the *R* value. Hence, an increase in *A* may result in a slight increase in the *B* value while simultaneously lowering the *R* value from 88% to 76%. In addition to high *A* and *R* values, an efficient PRO membrane should also possess a low *B* value.

### 3.3. Effects of the M_w_ of PEG on PWP and EWC

The results for *PWP* and *EWC* are presented in Figure 8. An increasing pattern of *PWP* was obtained when the M_w_ of PEG was increased. The relation between *EWC* and *PWP* plays a vital role in evaluating the hydrophilicity of the membranes [9]. The increased water permeability was observed to be due to the increased membrane pore sizes at higher M_w_s of PEG, ultimately leading to better pore formation and improved surface hydrophilicity. In other words, a higher M_w_ of PEG in the membrane matrix increased the flow rate of the water through the membrane [21,24]. Hence, as the M_w_ of PEG increased, the *PWP* and *EWC* increased as well. However, the *B* value increased, and the *R* value decreased (Figure 7), which means that the increased pore sizes in the membranes with higher M_w_ of PEG led to lower mechanical strengths and performances in the PRO mode, a finding that will be discussed further in the following section.

### 3.4. Effects of the M_w_ of PEG on Membrane Porosity and Hydrophilicity

Figure 9a,b shows the correlation between porosity, average pore size and the contact angle for all synthesized membranes. It can be seen from Figure 9a that the overall porosity and average pore size increased with increases in the M_w_ of the PEG. This finding can be explained by the slower phase inversion rate, which produced larger-sized finger-like pores [22]. Hence, an obvious increase in *PWP* was observed, as shown in Figure 8.

The contact angle (CA) results show a decreasing trend from 88.81° for pristine PSf to 61.57° for PA/PSf-6000. It was found that as the M_w_ of the PEG increases, the CA decreases, which indicates an increase in the surface hydrophilicity of the membrane, as shown in Figure 9b. The results indicate that the hydrophilicities of the TFC-1500, TFC-4000 and TFC-6000 membranes improved markedly. PEG with a higher M_w_ is more likely to be entrapped in the polymer matrix due to the slower phase-inversion rate in the coagulation bath, which altered the membrane’s characteristics in a good way, by improving its hydrophilicity [21]. Given that PEG is hydrophilic in nature, a higher M_w_ of the PEG may lead to more residuals of PEG, thus improving the hydrophilicity of membranes with low CA values. Hydrophilicity increases the interaction between the water molecules and pores on the polyamide surface area that influence water permeability, and hence, increase the water flux of the membrane. In addition, the finger-like pores of the modified membrane, shown in Figure 3, are asymmetric but hydrophilic, so water easily permeates to the pores of the membrane substrate [42].

The increasing formation of amide and –COOH groups in the PA layer (Figure 4 and Figure 5) influences the strength and repeated axes of the hydrogen bonds [43]. Zhang et al. [44] reported that the close interactions between water and the hydrophilic functional group in the PA layer improves the adsorption capacity of the membrane surface and could enhance the transport of water molecules through the membrane.

### 3.5. FO Membrane Performance and Structural Parameter

The performance of FO was assessed using 1 M NaCl as the DS, and DI water as the FS. The data for each membrane was collected for 5 h after the system was stabilized for 1 h. Figure 10 shows the water flux, *J_w_*, trends for the membranes with different M_w_s of PEG. For the initial 30 minutes of operation, it can be seen that TFC-6000 possessed the highest flux of 12.98 Lm^−2^h^−1^, followed by TFC-600 and TFC-4000 with fluxes of 12.40 Lm^−2^h^−1^ and 11.5 Lm^−2^h^−1^, respectively. However, over time, the *J_w_* for TFC-6000 showed a sharp decline, between approximately 30 and 80 minutes in, and started to stabilize at the 100th minute.

Unlike TFC-6000, a stable reading of TFC-600 membrane throughout the FO experiment was observed. The lowest *J_w_* was produced by TFC-400d due the smaller pore sizes in the membrane, which resulted in a severe dilutive ICP that potentially reduced the osmotic driving force across the FO membrane [45].

On the other hand, the high *J_w_* obtained when TFC-600 was used could be related to its relatively low reverse salt flux, *J_s_*, compared to those for TFC-4000 and TFC-6000 (Figure 11a). The trend of *J_s_* has been found to be consistent with the *R* value. The *J_s_* value increases with increases in the M_w_ of PEG, due to the reduced surface roughness of the crosslinked PA active layer, for higher M_w_s of PEG membranes. This reduces the surface roughness, which ultimately reduces the salt rejection efficiency and increases the *J_s_* [22].

Figure 11 shows the specific salt fluxes, *J_s_*/*J_w_*, which were estimated and used to determine the osmotic process efficiencies and compare the performances of the modified TFC membranes with different M_w_s of PEG. The best FO membrane must possess a high *J_w_* and a low *J_s_*; thus, membranes with a low *J_s_*/*J_w_* are preferred [46]. From Figure 11a, it can be seen that the *J_s_*/*J_w_* increased significantly with the increase in the M_w_ of the PEG. When taking into consideration all the measured and analysed data, TFC-600 was found to meet all the criteria required compared to the other synthesized membranes in this study.

The structural parameter, *S*, can be expressed as the depth of the solutes diffusing across the substrate layer, and it is an important parameter to evaluate when examining the ICP effect in the FO process. Figure 11b reveals that the pristine TFC-PSf membrane had the highest *S* value at 1394 μm. The TFC-600 had the lowest *S* value at 811 μm, which is consistent with the *J_w_* and *J_s_* data. *A* low *S* value is likely to be preferred for osmotically driven membranes due to its potential to have less of an ICP effect [12,38].

### 3.6. PRO Membrane Performance

In PRO, the power density, *W*, is determined by the product of water flux, *J_w_*, and pressure difference, Δ*P*, as indicated in Equation (9). In this study, the TFC PA/PSf-PEG membranes with different M_w_s of PEG were evaluated using 1 M NaCl as the DS, and DI water as the FS, with the active layer facing the DS (AL-DS). All the membranes had good mechanical strength, as they could withstand pressures of up to 8 bar.

Figure 12 shows the trends of the power density achieved by the membranes over a 5-hour period under a pressure of >10 bar. All the data were collected for 1 h for each pressure after 30 min stabilization, then the pressure was increased. The figure consists of modeled (dots) and experimental (symbol) data for all the membranes. All the membranes’ experimental data followed the PRO model until they reached bursting point at >10 bar. At a pressure of 8 bar, the TFC-600 membrane reached a peak, *W*, of 2.22 Wm^2^. This result was expected, as TFC-600 had the highest *J_w_* of 10.10 Lm^−2^h^−1^ and a sufficiently low *J_s_* (Figure 13) compared to the other fabricated membranes, due to its porosity and hydrophilicity, which plays a vital role in the transportation of salt and water molecules.

Meanwhile, the *W* for TFC-6000, which had the highest *A* value, only generated an average *W* of 0.3 Wm^−2^ throughout the entire 5-hour period. The other membranes suffered more severe ICP and delivered lower performances compared to TFC-600. Figure 13 shows that the value of *J_w_* sharply decreased from TFC-1000 to TFC-6000, while *J_s_* increased and eventually affected the *W* produced. The low *W* produced by higher M_w_s of PEG significantly impacted the membrane structures with larger pore sizes, as these pores may have been deformed due to the compression, elongation and bending stresses resulting from the application of pressure [47].

## 4. Conclusions

In this work, modified TFC membranes with different M_w_s of PEG were synthesized for FO and PRO applications. The effects of PEG additions with different M_w_s on the PSf substrate properties were investigated. In general, the SEM images showed that the morphologies of the membranes were very loose, with larger finger-like pores emerging for higher M_w_s of PEG. As the M_w_ of the PEG increased, the porosity, average pore size and hydrophilicity increased significantly. In this study, the main reason for the performance improvement of the prepared TFC membrane was the formation the substrate with fine pore structure or the formation of a hydrophilic substrate. In addition, it was discovered that membranes with a higher M_w_ of PEG had lower surface roughness when a PA layer was incorporated onto the membrane top layer, which significantly increased *J_s_* and lowered the membrane performance in the FO/PRO process. In both the FO and PRO modes, the results indicate that the TFC-600 membrane has the preferred characteristic over the other synthesized membranes. The TFC-600 membrane achieved a *J_w_* value of 12.40 Lm^−2^h^−1^. Furthermore, an *S* value of 811 μm was obtained in the FO mode, and a power density of 2.2 Wm^−2^ was obtained in the PRO mode with 1 M NaCl, and with DI water as the draw solution and feed solution, respectively. Future research should further develop and confirm these findings by studying the optimal performance of TFC membranes, by further exploring the chemical structure of the additive and the membrane substrate.

## Figures and Tables

**Figure 1 membranes-12-00282-f001:**
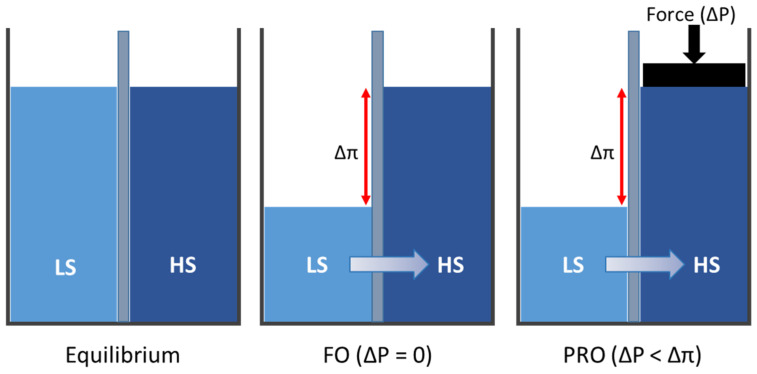
Comparison of FO and PRO processes.

**Figure 2 membranes-12-00282-f002:**
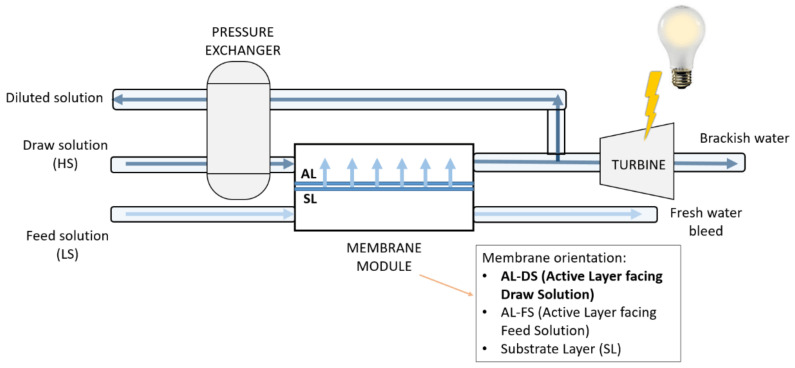
PRO system.

**Figure 3 membranes-12-00282-f003:**
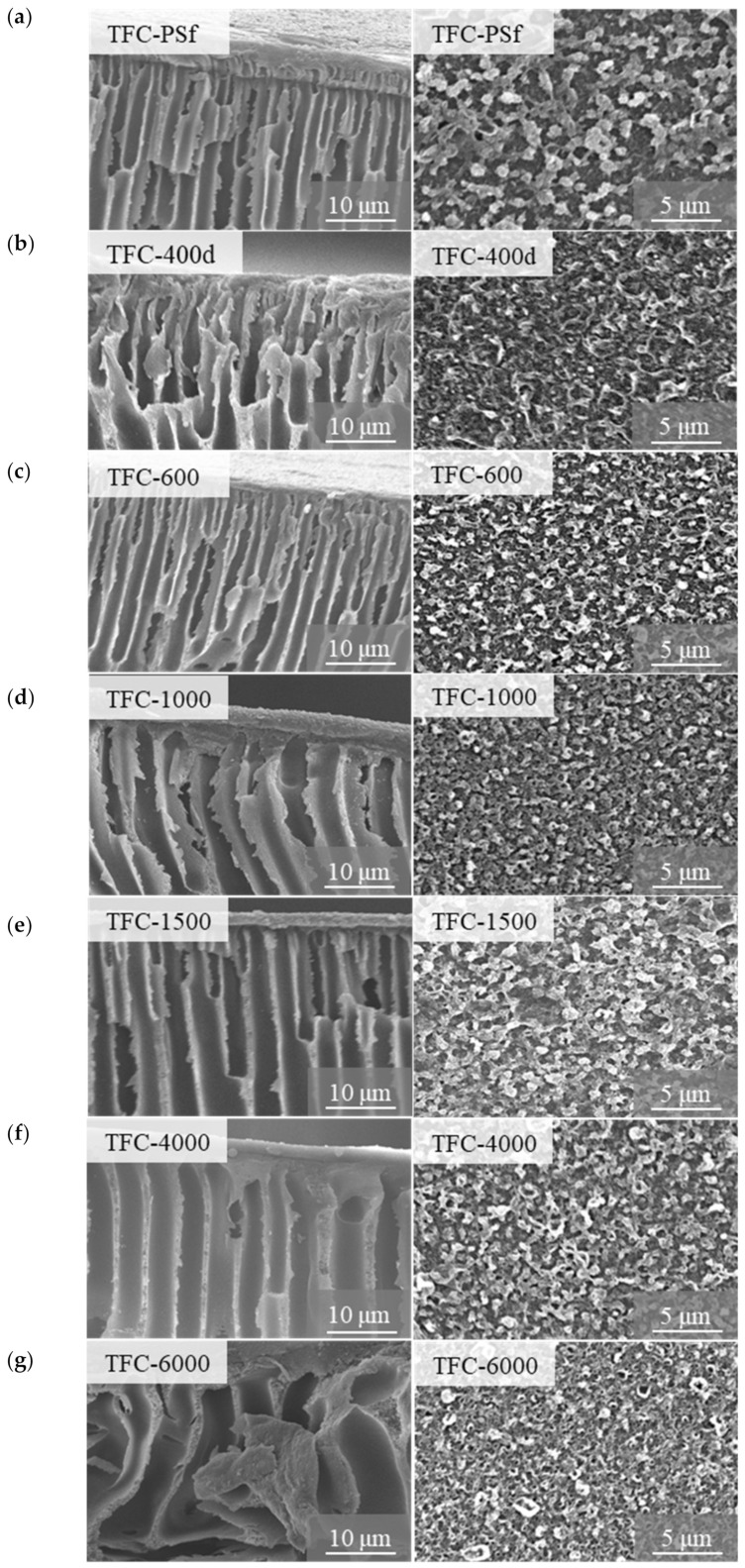
SEM images of the cross-sections and top surfaces of the pristine TFC membrane and modified TFC membranes: (**a**) TFC-PSf; (**b**) TFC-400d; (**c**) TFC-600; (**d**) TFC-1000; (**e**) TFC-1500; (**f**) TFC-4000; and (**g**) TFC-6000.

**Figure 4 membranes-12-00282-f004:**
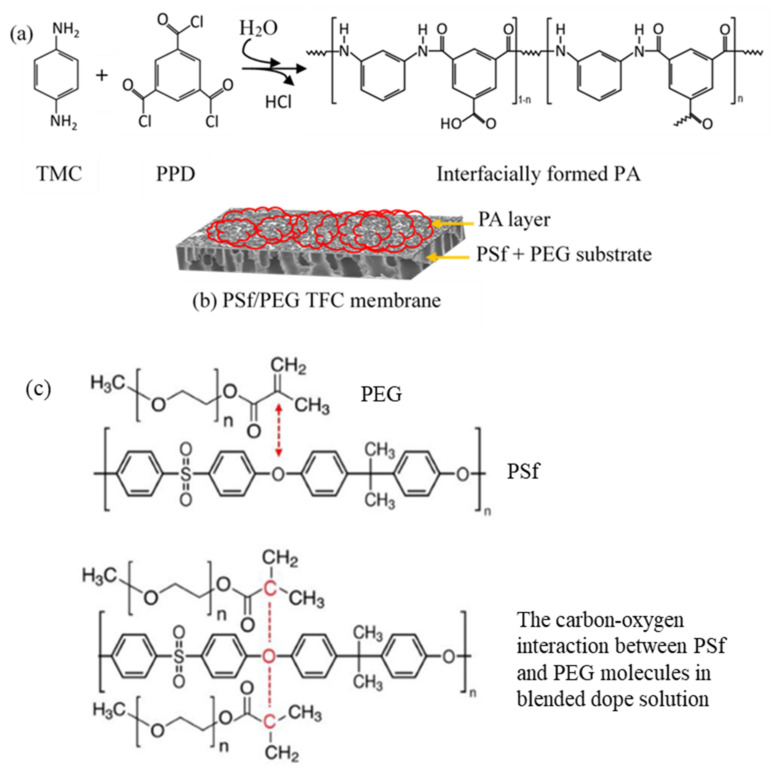
(**a**) Crosslinking reaction of the PA layer on the TFN membrane; (**b**) modified TFC membrane; and (**c**) synthesis pathway for the modified membrane substrate with PEG [28].

**Figure 5 membranes-12-00282-f005:**
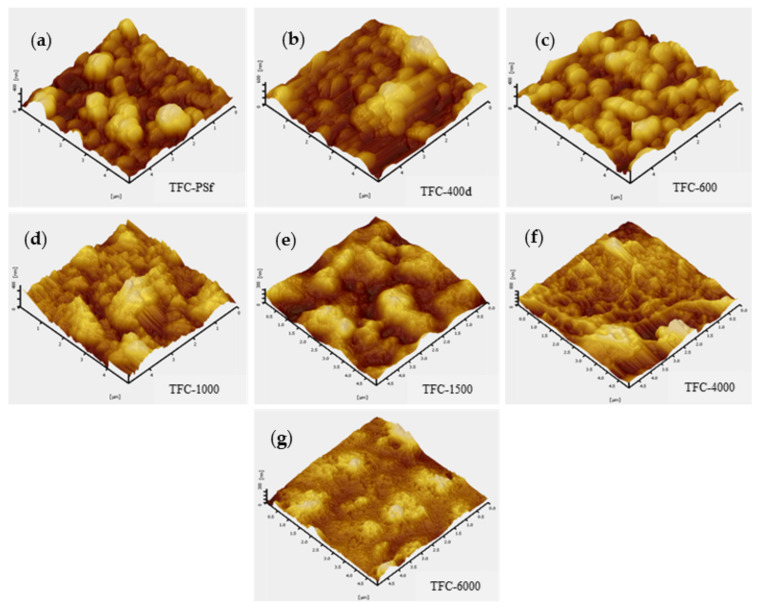
AFM 3D images of the pristine TFC membrane and modified TFC membranes: (**a**) TFC-PSf; (**b**) TFC-400d; (**c**) TFC-600; (**d**) TFC-1000; (**e**) TFC-1500; (**f**) TFC-4000; and (**g**) TFC-6000.

**Figure 6 membranes-12-00282-f006:**
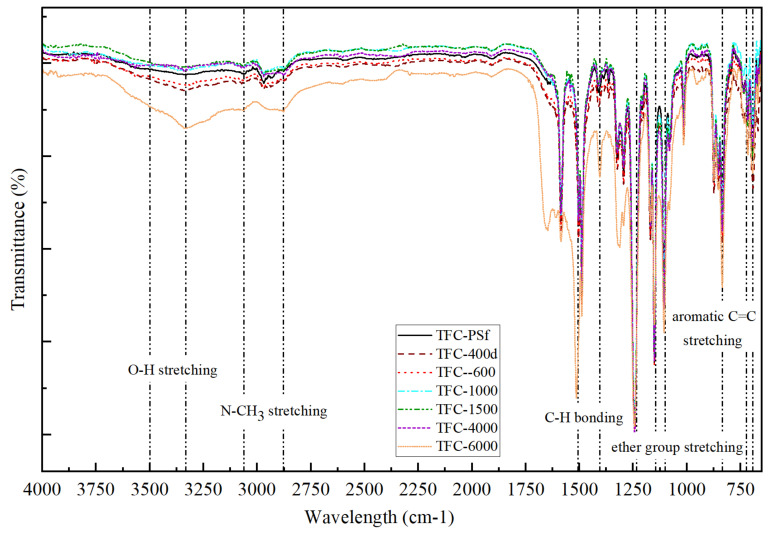
FTIR spectra of the pristine TFC-PSf membrane and modified TFC membranes with different M_w_ of PEG.

**Figure 7 membranes-12-00282-f007:**
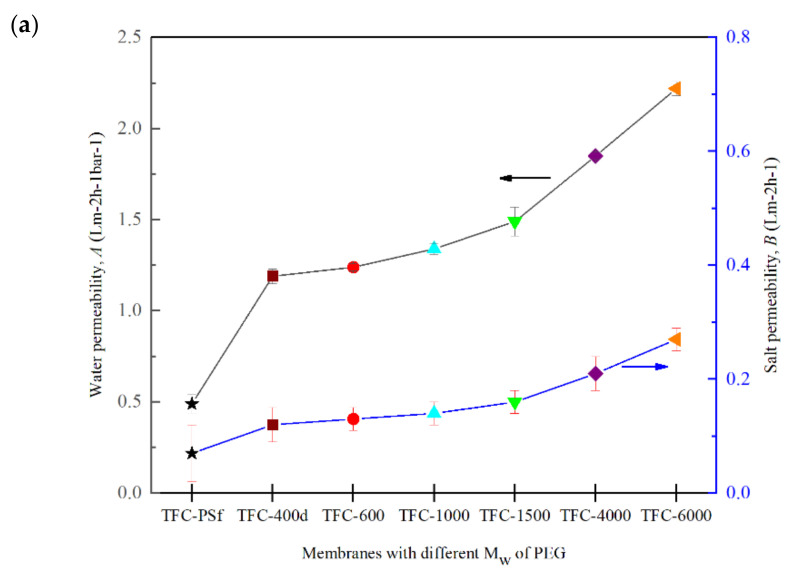
(**a**) Water permeability, *A*, and salt permeability, *B*; (**b**) salt rejection, *R*, for modified TFC membranes with different M_w_s of PEG.

**Figure 8 membranes-12-00282-f008:**
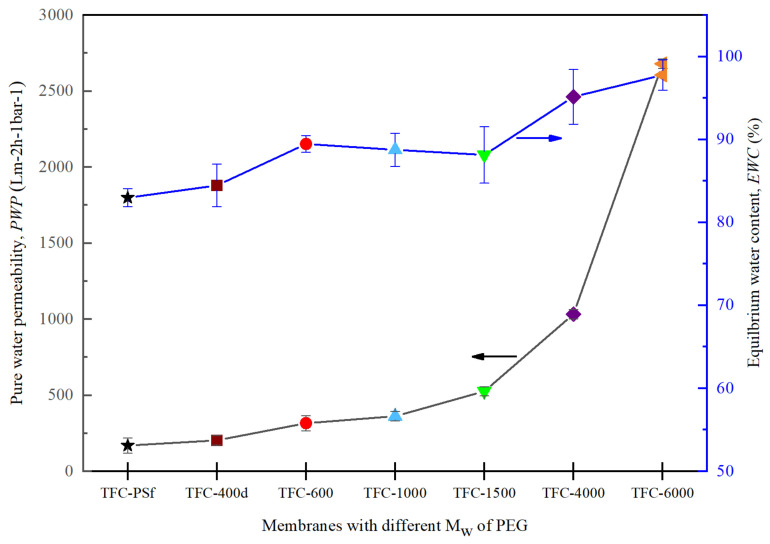
Pure water permeability, *PWP*, and equilibrium water content, *EWC*, of the modified TFC membranes with different M_w_s of PEG.

**Figure 9 membranes-12-00282-f009:**
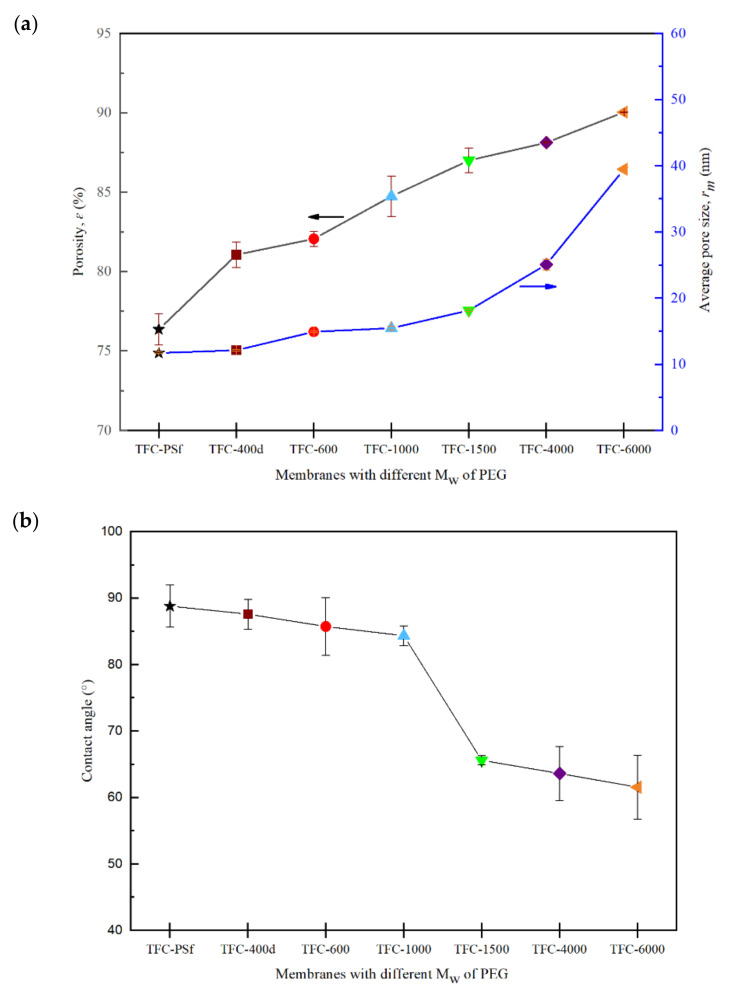
(**a**) Porosity, *ε* and average pore size, *r_m_*, and (**b**) contact angle of the modified TFC membranes with different M_w_ of PEG.

**Figure 10 membranes-12-00282-f010:**
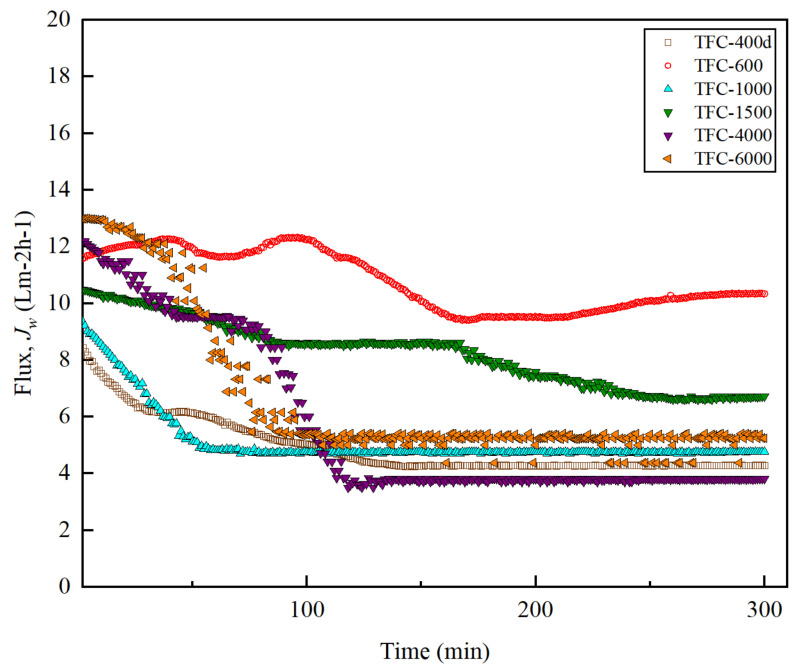
Water flux (*J_w_*) for the modified TFC membranes with different M_w_s of PEG over operating time (min) in FO mode.

**Figure 11 membranes-12-00282-f011:**
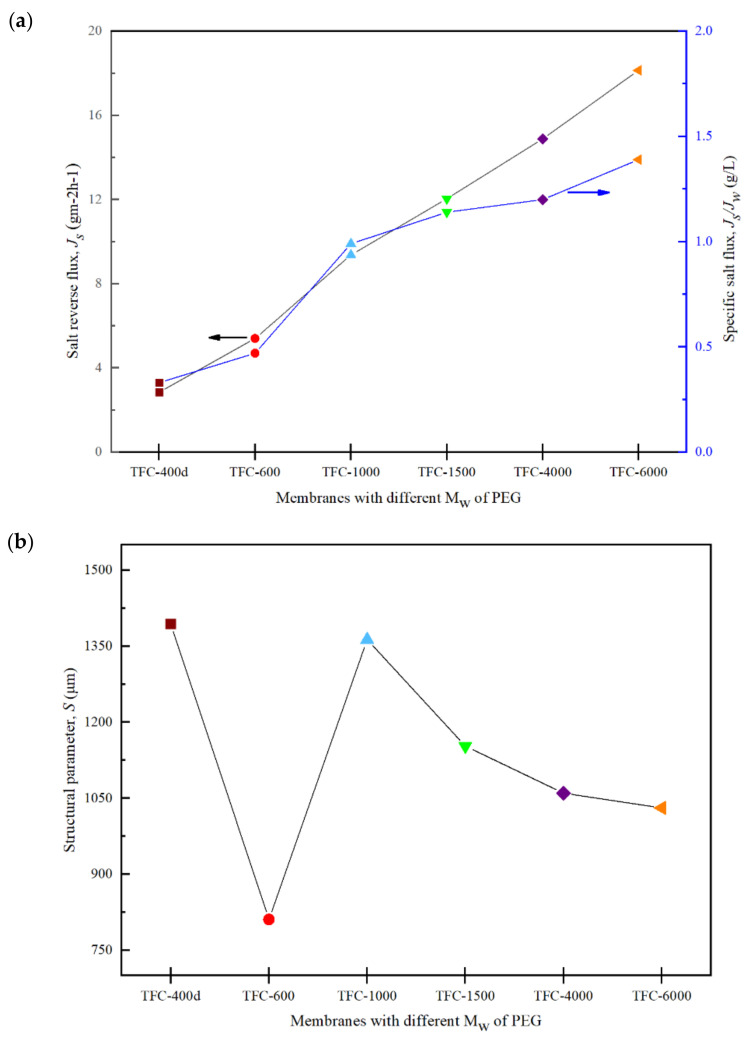
(**a**) Salt reverse-flux, *J_s_*, and specific salt flux, *J_s_*/*J_w_*; and (**b**) structural parameter, *S*, of the modified TFC membranes with different M_w_s of PEG in FO mode.

**Figure 12 membranes-12-00282-f012:**
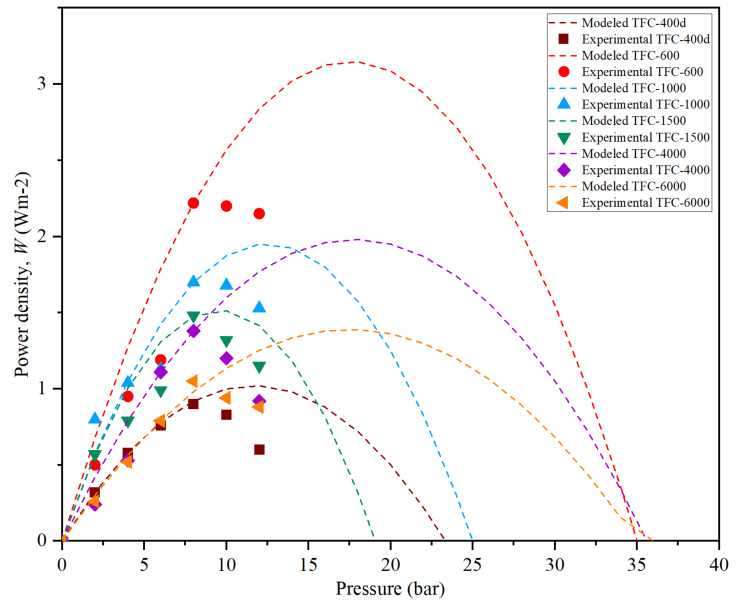
The modeled and experimental results on the power-density profiles for modified TFC membranes with different M_w_s of PEG.

**Figure 13 membranes-12-00282-f013:**
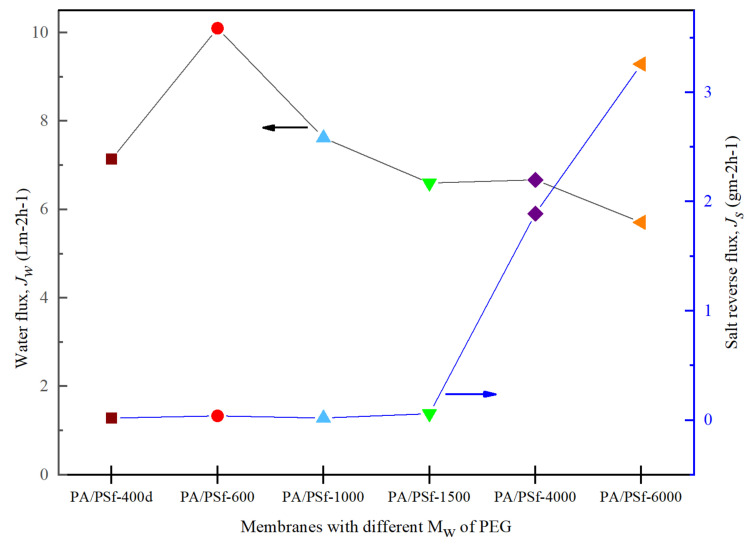
The water flux, *J_w_*, and salt-reverse flux, *J_s_*, in the PRO experiment for modified TFC membranes with different M_w_s of PEG.

**Table 1 membranes-12-00282-t001:** Composition of casting solutions used in the fabrication of membranes’ substrates.

Membrane	Mw of PEG (gmol^−1^)	PEG/NMP/PSf (wt%)
PSf	0	0/82/18
PSf-400a	400	0.25/81.75/18
PSf-400b	400	0.5/81.50/18
PSf-400c	400	0.75/81.25/18
PSf-400d	400	10/82/18
PSf-600	600	10/82/18
PSf-1000	1000	10/82/18
PSf-1500	1500	10/82/18
PSf-4000	4000	10/82/18
PSf-6000	6000	10/82/18

**Table 2 membranes-12-00282-t002:** Maximum peak-to-valley distances, average surface roughness and root-mean-squared roughness for the TFC PA/PSf-PEG membranes with different M_w_ of PEG.

Membrane	Maximum Peak-to-Valley Distance, *R_p-v_* (μm)	Surface Roughness (μm)	
		Average, *R_avg_*	Root-Mean-Squared, *R_rms_*
TFC-PSf	54.63	84.83	10.75
TFC-400d	60.48	73.54	94.37
TFC-600	62.43	68.11	86.49
TFC-1000	53.89	62.26	78.32
TFC-1500	32.64	43.44	53.42
TFC-4000	41.05	38.89	51.93
TFC-6000	37.17	31.42	41.64

**Table 3 membranes-12-00282-t003:** Intrinsic properties of the pristine TFC-PSf membrane and modified TFC membranes with PEG 400.

TFC Membrane	Water Permeability, *A* (Lm^−2^h^−1^bar^−1^)	Salt Permeability, *B* (Lm^−2^h^−1^)	Salt Rejection, *R* (%)	Structural Parameter, *S* (μm)
TFC-PSf	0.49 ± 0.12	0.07 ± 0.26	68 ± 0.29	1803
TFC-400a	0.89 ± 0.02	0.11 ± 0.04	71 ± 1.01	1414
TFC-400b	1.01 ± 0.01	0.12 ± 0.05	79 ± 0.50	1407
TFC-400c	1.14 ± 0.01	0.13 ± 0.09	82 ± 0.87	1399
TFC-400d	1.19 ± 0.04	0.16 ± 0.05	88 ± 0.37	1394

**Table 4 membranes-12-00282-t004:** The *A*, *B* and *R* values of the TFC-600 membrane and other TFC membranes with PEG 600 found in the literature.

Membranes	Water Permeability, *A* (Lm^−2^h^−1^bar^−1^)	Salt Permeability, *B* (Lm^−2^h^−1^)	Salt Rejection, *R* (%)	Ref.
TFC-600	1.24 ± 0.02	0.13 ± 0.02	88 ± 0.43	This study
MS3(PSf/PEG-600/NMP) ^a^	1.37 ± 0.21	1.89 ± 0.29	82 ± 1.52	[38]
TFN-0.1GO (PSf/PEG-600/GO/NMP) ^b^	1.04 ± 0.33	0.37 ± 0.02	96.2 ± 0.08	[39]
PSf/PEG-600/Tween80/ * DMF	n/a	n/a	83.2	[40]
** CA/PEG-600 (10 g) ^c^	0.27	n/a	52	[41]
CA/PEG-600 (8 g) ^c^	0.25	n/a	60	
CA/PEG-600 (6 g) ^c^	0.20	n/a	69.2	
CA/PEG-600 (4 g) ^c^	1.68	n/a	81.5	

* N,N-dimethyl formamide, ** Cellulose acetate. ^a^ 2 gL^−1^ NaCl solution at 10–18 bar; ^b^ 1000 ppm of NaCl at 10 bar; ^c^ NaCl (10 mM) aqueous solutions under an operating pressure of 0.5 MPa.

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
