# Peer review of "Modification of Thin Film Composite Pressure Retarded Osmosis Membrane by Polyethylene Glycol with Different Molecular Weights"

_membranes, 2022, doi:10.3390/membranes12030282_

Round 1

Reviewer 1 Report

In this current manuscript, TFC PRO membrane was prepared by the authors by using PEG modified PES substrate. The effect of the molecular weight of PEG on the performance of the TFC membranes was investigated. However, this manuscript is poorly written and organized. Some discussions and conclusions presented in this work are questionable. Therefore, in this current situation, this manuscript is not suggested to be published in Membranes. More comments can be found below.

  1. The introduction part is suggested to be well-organized to show the novelty of this work.
  2. The influence of structure and properties of the substrate on the formation and performance of the TFC membranes has been reported by many researchers. To clearly evaluated the effect of the molecular weight of PEG on the performance of the TFC membranes, the structure and properties of the substrate, such as the surface pore size, porosity and surface hydrophilicity, should be characterized.
  3. As reported by many researchers, PEGs were always used as the pore-forming agents in the preparation of porous membranes. While PEGs were also reported as the agents for the modification of membrane hydrophilicity. For FO or PRO membrane, membrane performance was greatly influenced by the pore structure and hydrophilicity of the substrate. Thus, in this study, the main reason for the performance improvement of the prepared TFC membrane was the formation the substrate with fine pore structure or the formation of a hydrophilic substrate.
  4. In figure 4 c, the author presents the chemical bonding structure of PSf and PEG. This reaction was highly questionable, please provide some evident to prove it.
  5. The salt rejection properties were closely relative to the cross-link degree of the PA in the rejection layer. The cross-link degree information is suggested to be provide for the better explanation for the variation of TFC membrane performance.
  6. Figure 6 are not clearly and suggested to be refined.
  7. The captions and figure of Fig.8, Fig.9 and Fig. 10 were mismatched.

Reviewer 2 Report

This manuscript study the effect of PEG Mw on properties and FO/PRO performance of thin film composit membrane. The manascript is well organised and written. However, it can be improved  before accepted to publish as follow;

  1. Intorduction section,                                                                            ----Figure 2, replace PRO pilot plant with, Pro system                                  -In addtion to PEG,  few additive materials have been used as a promissing additive to improved for FO/PRO system (e.g. graphene). Please provide this imformatiom  in the introduction section                 - Please state the hyphothesis or  research question of this study 
  2. Result and dicussion                                                                                   - Page 15, In addtion to pore structure/size, please explain the role of hydrophilicities of the sopport layer on water flux or ICP.                            - Page15,  Regarding the Js or salt rejection of TFC-600 and TFC-6000, Is it possible that the difference in flux behavior was influenced by   Js/ECP?                                                                                               -Please suggest the future work may be required.

Round 2

Reviewer 1 Report

In this revision, parts of the comments have been responded. The current version of this manuscript is suggested to be published in Membranes after minor revision.

Some comments are listed for further improving the current work:

1. To clearly evaluated the effect of the molecular weight of PEG on the performance of the TFC membranes, the structure and properties of the substrate, such as the surface pore size, porosity and surface hydrophilicity, should be characterized. In this revision, the author only provided the information of the prepared TFC membrane. What about the substrates?

2. In this revision, Figure 4 had been revised. However, this revised chemical bonding structure of PSf and PEG (the carbon-oxygen interaction between PSf and PEG) is still questionable. To avoid the misleading to the readers, the author must provide some solid evidence (for example, NMR).
